# Laboratory evaluation of twelve portable devices for medicine quality screening

Stephen C. Zambrzycki[1‡], Celine Caillet[2,3,4‡], Serena Vickers[2,3,4], Marcos Bouza[1], David V. Donndelinger[1], Laura C. Geben[1], Matthew C. Bernier[1], Paul N. Newton[2,3,4], Facundo M. Fernández[1]*

1 School of Chemistry and Biochemistry, Georgia Institute of Technology, Atlanta, Georgia, United States of America, 2 Lao-Oxford-Mahosot Hospital-Wellcome Trust-Research Unit, Microbiology Laboratory, Mahosot Hospital, Vientiane, Lao PDR, 3 Centre for Tropical Medicine and Global Health, Nuffield Department of Medicine, University of Oxford, United Kingdom, 4 Infectious Diseases Data Observatory (IDDO) & WorldWide Antimalarial Resistance Network (WWARN), University of Oxford, Oxford, United Kingdom

‡ These authors share first authorship on this work.
* facundo.fernandez@chemistry.gatech.edu

**Data Availability Statement:** All relevant data are within the manuscript and its Supporting information files.

## Abstract

### Background

Post-market surveillance is a key regulatory function to prevent substandard and falsified (SF) medicines from being consumed by patients. Field deployable technologies offer the potential for rapid objective screening for SF medicines.

### Methods and findings

We evaluated twelve devices: three near infrared spectrometers (MicroPHAZIR RX, NIR-S-G1, Neospectra 2.5), two Raman spectrometers (Progeny, TruScan RM), one mid-infrared spectrometer (4500a), one disposable colorimetric assay (Paper Analytical Devices, PAD), one disposable immunoassay (Rapid Diagnostic Test, RDT), one portable liquid chromatograph (C-Vue), one microfluidic system (PharmaChk), one mass spectrometer (QDa), and one thin layer chromatography kit (GPHF-Minilab). Each device was tested with a series of field collected medicines (FCM) along with simulated medicines (SIM) formulated in a laboratory. The FCM and SIM ranged from samples with good quality active pharmaceutical ingredient (API) concentrations, reduced concentrations of API (80% and 50% of the API), no API, and the wrong API. All the devices had high sensitivities (91.5 to 100.0%) detecting medicines with no API or the wrong API. However, the sensitivities of each device towards samples with 50% and 80% API varied greatly, from 0% to 100%. The infrared and Raman spectrometers had variable sensitivities for detecting samples with 50% and 80% API (from 5.6% to 50.0%). The devices with the ability to quantitate API (C-Vue, PharmaChk, QDa) had sensitivities ranging from 91.7% to 100% to detect all poor quality samples. The specificity was lower for the quantitative C-Vue, PharmaChk, & QDa (50.0% to 91.7%) than for all the other devices in this study (95.5% to 100%).

**Funding:** This work has been co-funded by the Regional Malaria and other Communicable Disease Threats Trust Fund, which has been co-financed by the Government of Australia (Department of Foreign Affairs and Trade); the Government of Canada (Department of Foreign Affairs, Trade and Development); and the Government of the United Kingdom (Department for International Development). The grant ( RETA 8763 ) was managed by the Asian Development Bank (https://www.adb.org/) and awarded to PNN. Additional support was provided by Wellcome Trust Grant N° 202935/Z/16/Z (https://wellcome.org/). The funders had no role in study design, data collection and analysis, decision to publish, or preparation of the manuscript. For the purpose of Open Access, the author has applied a CC BY public copyright licence to any Author Accepted Manuscript version arising from this submission.

**Competing interests:** The authors have declared that no competing interests exist.

## Conclusions

The twelve devices evaluated could detect medicines with the wrong or none of the APIs, consistent with falsified medicines, with high accuracy. However, API quantitation to detect formulations similar to those commonly found in substandards proved more difficult, requiring further technological innovation.

## Author summary

Criminally falsified or poorly manufactured medicines can lead to patients becoming sicker and losing trust in the health system. Portable tools beyond just documentation checks and visual inspection (the current practices in many low- and middle-income countries) can help pharmacy inspectors with early detection of poor quality medicines. Currently, many tools are available to detect poor quality medicines, but their performances have not been properly assessed and compared. In this study, 12 different devices ranging from disposable single use tests to portable spectrometers were tested in a laboratory. All the tested devices could identify medicines that contained none or the wrong active ingredient(s), a common trait of falsified medicines. Disposable tests required few resources to be implemented, but had difficulties identifying medicines with reduced amounts of active ingredients. Spectrometers used 'out-of-the-box' required minimal consumables had varying degrees of success at detecting medicines with reduced amounts of active ingredients. Finally, instruments with more quantitative abilities, such as benchtop simple chromatographs or mass spectrometers, offered the best sensitivity for detecting medicines with reduced amounts of active ingredients, but required the most resources and training and were deemed to be more suitable for centralized testing.

## Introduction

Poor quality medicines are classified by the World Health Organization (WHO) as substandard or falsified (SF)[1]. Falsified medicines purport to be real, authorized medicines, but they 'deliberately/fraudulently misrepresent their identity, composition or source'[1]. Falsified medicines are the result of criminal activity and usually have packaging that is a copy of a genuine product. They may contain the incorrect amount of the correct active pharmaceutical ingredient (API), wrong API(s), or more commonly, no API at all. Substandard medicines are those authorized medical products 'that fail to meet either their quality standards or their specifications, or both' [1]. They negate the benefits of modern medicines, failed treatments, increased antimicrobial resistance, and distrust in the health system [2].

The recent expansion of the diversity of portable devices for medicine quality screening holds great hope for empowering inspectors in the field, making their work more cost-effective and actionable, improving medicine regulatory agency capacity, and protecting patients from the harms of SF medicines. However, significant knowledge gaps exist regarding the optimal choice of device, or combination of devices. These gaps impede decisions on how to best use these portable devices [3,4].

As described in the first paper in this Collection 'A multi-phase evaluation of portable screening devices to assess medicines quality for national Medicines Regulatory Authorities', we undertook a multi-phase collaborative project to evaluate diverse devices for SF medicine

detection, producing key information required for deciding which of these technologies may be the most appropriate for field medicine quality screening.

In this second paper we present the results for the evaluation of 12 portable screening technologies for distinguishing between genuine good quality, 50% and 80% API medicines (mimicking substandard medicines), and 0% and wrong API medicines (mimicking falsified products) under controlled laboratory conditions. We present protocols for utilizing each device, an evaluation of the devices' ease of operation and diagnostic accuracy, and an assessment of the devices' resource needs. We close with suggestions of which devices were prioritized for field testing, described in the third paper of the series.

## Methods and materials

### Devices

Following a detailed literature review [3] and further discussions with experts, twelve devices were chosen for evaluation (Table 1). These included single-use Paper Analytical Devices (PAD), Rapid Diagnostic Tests (RDT), a portable liquid chromatograph, a single quadrupole benchtop mass spectrometer, the PharmaChk microfluidic device, an attenuated total reflectance mid-IR (MIR) spectrometer, three different near-infrared (NIR) spectrometers, two

**Table 1. List of instruments and devices evaluated, including their underlying technology and basic instrument specifications.**

| Technology | Device Name (Manufacturer) | Basic Instrument Specifications | APIs Tested | Market Status at the Time of this Study |
|---|---|---|---|---|
| Colorimetric Assay | Paper Analytical Devices (PAD, Notre Dame University). | 12 colorimetric chemical tests on a single-use paper card. | A, AZITH, P, OFLO, & SM | Under Development [5–7] |
| Lateral Flow Immunoassay | Rapid Diagnostic Tests, (RDT, China Agricultural University of Beijing and University of Pennsylvania). | Single-use disposable dipsticks, concentration specific. | Artemether, ART, & DHA | Under Development [8–10] |
| Liquid Chromatography | C-Vue (C-Vue). | Mercury lamp ($\approx$254 nm) detector; Millipore Chromolith RP18e 25 x 4.6 mm column. | ACA, OFLO, & SMTM | Marketed [11] |
| Mass Spectrometry | QDa (Waters). | Selected ion monitoring mode per API; flow injection analysis. | All | Marketed [12] |
| Microfluidics | PharmaChk (Boston University). | Luminescence detection, 490 & 515 nm LED & filters. | ART | Under Development [13] |
| Mid-Infrared Spectroscopy | 4500a FTIR Single Reflection[†] (Agilent) | Attenuated Total Reflectance. Spectral range: 2,500–15,384 nm. | All | Marketed [14] |
| Near-Infrared Spectroscopy | NeoSpectra 2.5[*] (Si-Ware). | Spectral range: 1,350–2,500 nm. | All | Marketed [15] |
| | MicroPHAZIR RX[*†] (Thermo Fisher Scientific). | Spectral range: 1,600–2,400 nm. | All | Marketed [16] |
| | NIR-S-G1[*†˚] (Innospectra). | Spectral range: 900–1,700 nm. | All | Marketed [17, 18] |
| Raman Spectroscopy | Progeny[*†] (Rigaku). | Excitation Laser: 1,064 nm, Spectral range 200–2,500 cm$^{-1}$. | All | Marketed [19] |
| | TruScan RM[*†] (Thermo Fisher Scientific). | Excitation Laser: 785 nm, Spectral range 250–2,875 cm$^{-1}$. | All | Marketed [20] |
| Thin Layer Chromatography | Minilab[‡] (Global Pharma Health Fund E.V.). | Detection by chemical staining and UV light exposure. | All | Marketed [21] |

A: Amoxicillin; API, Active pharmaceutical ingredient; ART: Artesunate; AZITH: Azithromycin; CA: Clavulanic Acid; DHA: Dihydroartemisinin; FTIR: Fourier Transform Infrared, OLFO: Ofloxacin; P: Piperaquine; SM: Sulfamethoxazole; TM: Trimethoprim.

[*]Instrument/device could also scan samples through transparent packaging.

[†]Automatic spectral reference library comparison capabilities.

[‡]Only TLC portion of the Minilab was used in this study, but not the dissolution and weighing tests.

[˚]Also referred to in this study as, "NIRscan", the beta version used in this study.

Raman spectrometers, and the Minilab thin layer chromatography kit. Brief fundamentals of each underlying technology are explained in S1 Appendix.

These 12 devices were chosen to represent a variety of different technologies within the project's budget and time constraints. The ideal device should be portable, battery powered, durable, and require minimal training, consumables, sample preparation, and test the APIs included in this study. The two Raman spectrometers (TruScan RM and Progeny) were chosen for their different laser excitation sources. The infrared spectrometers (4500a, MicroPHAZIR RX, Neospectra 2.5, NIR-S-G1) were selected because they each had different testing modalities along with sampling in different infrared spectral ranges. The remaining devices were selected because they are representative of key technologies used in the screening of small molecule medicines. Key configuration exceptions and constraints are described in S2 Appendix. Photographs of each device are in S3 Appendix. Detailed descriptions of each of these devices, including a more detailed overview of the technology, initial purchase price, and device specifications are provided in S4 Appendix.

The device specific operating protocols developed for this project, sample preparation conditions, and instrument parameters are detailed in S5 Appendix.

## Samples

Antibiotic and antimalarial formulations represent a majority of the total number of SF medicines reported to the WHO Global Surveillance and Monitoring System between 2013–2017 [22]. Seven key antibiotics and antimalarials used in South-East Asia were chosen for device evaluation: AL = Artemether-Lumefantrine; ART = Artesunate; AZITH = Azithromycin; ACA = Amoxicillin-Clavulanic Acid; DHAP = Dihydroartemisinin-Piperaquine; OLFO = Ofloxacin; and SMTM = Sulfamethoxazole-Trimethoprim.

Field-collected medicines (FCM) and simulated medicines (SIM) were used to test the devices. FCM included good quality, falsified, and "look-alike" medicines. All FCM samples were tested with ultra-performance liquid chromatography (UPLC) or mass spectrometry (MS) to ensure quality (S6 Appendix). Good quality FCM were purchased from reliable wholesalers in the Greater Mekong Sub-Region or donated by manufacturers. Falsified FCM were collected during earlier studies (see [23] and references therein). Look-alike FCM were visually indistinguishable from genuine medicines, and contained APIs not evaluated in this study [24]. Five brands of FCM were sold in transparent packaging such as clear plastic blister packs or clear glass vials. Five devices tested claimed to be capable of performing nondestructive testing through transparent barriers (Table 1). Whenever feasible, FCM were also evaluated while sealed in their original packaging. Field-collected parenteral artesunate (Artesun) powder was removed from the vial and transferred to a plastic re-sealable bag for testing with Raman instruments because the powder was too dispersed in the original vial for one of the spectrometers to yield a stable signal and a consistent result. Device operators were not blinded to the identity of the samples being tested.

Most portable devices are only stated as able to check the presence/absence of the API or whole formulation spectra. Therefore, we did not examine dissolution rates, homogeneity of the API distribution within tablets, or tablet coating thicknesses. SIM were produced in the laboratory to mimic good quality, 80% and 50% API concentration tablets, mimicking substandard medicines, and no API and wrong API tablets, mimicking falsified medicines. All SIM were prepared as 100 mg 6 mm-diameter tablets, except for ART which was tested as a loose powder to simulate Artesun. ART, AZITH, OLFO, and SMTM powders (>98% purity) were purchased from TCI Chemical (Portland, OR). The simulated ACA, AL, and DHAP medicines, were produced from good quality FCM (AMK 1000 mg for ACA, Coartem for AL,

and D-Artepp for DHAP) that were crushed, mixed, and re-pressed into SIM tablets. The excipients included cellulose, lactose, or starch as the bulking agents, and magnesium stearate as the lubricant, all sourced from Sigma Aldrich (St. Louis, MO). The only ingredient in FCM genuine artesunate intravenous vials was artesunate. Magnesium stearate was thus excluded from the ART SIM formulation to simulate a loose powder intravenous artesunate. To mimic 80% and 50% substandards, the excipients listed above were added to dilute the API. Acetaminophen was used as the wrong API for falsified SIM and was sourced from Sigma Aldrich (St. Louis, MO). Details on specific formulations and methods for making each SIM are found in S7 Appendix, and protocols describing sample handling for each instrument/device are found in S8 Appendix.

## Data analysis

When testing medicine samples, devices generated results that were either qualitative or quantitative. Qualitative results were based upon pattern comparison between a known good quality medicine reference and the test medicine sample data. PAD chemically reacted with the medicine ingredients generating a color pattern, that was then visually compared to a reference photograph to determine the presence of an API. The 4500a, MicroPHAZIR RX, NIR-S-G1, Progeny, and TruScan RM spectrometers computationally compared experimentally-collected spectra to reference spectra of good quality medicines stored in the device's database. Each sample spectrum acquired was given a score by the device software resulting from the comparison with the good quality reference medicine spectrum. Such scores had to meet a given threshold to determine if a medicine passed (Fig 1A). Reference library creation is described in S9 Appendix. For the NIR-S-G1 spectrometer, reference samples were sent to the developer who prepared the reference libraries. The passing threshold values for the correlation coefficient or p-value testing initially set as default by the developer in the MicroPHAZIR RX, NIR-S-G1, Progeny, and TruScan RM spectrometers were utilized. These devices would directly tell the user 'pass' or 'fail', which were recorded. The pass threshold for the 4500a MIR spectrometer's correlation coefficient was set by us at >0.9 because the device would not output a direct pass/fail result, but rather give a list of matches with their associated correlation coefficients. The Neospectra 2.5 spectrometer did not include software to compare the experimentally-collected spectra to reference spectra. Therefore, experimental Neospectra 2.5 spectra were overlaid with reference spectra and visually compared by an analyst blinded to sample identity for determining the final pass/fail result.

To estimate the amount of API with quantitative instruments (C-Vue, PharmaChk and QDa) a calibration curve approach was used (Fig 1B). After sample preparation and data acquisition with the instruments, the calculated API amount had to be within ± 10% of the stated API amount for the sample to be classified as good quality, for both single and combination API medicines. The reference ranges of percent API(s) vary according to different pharmacopeias and different APIs (see S10 Appendix). For simplicity, we considered that medicines with %API outside the 90–110% range of the manufacturer's stated amount of API(s) were out of specification for any API included. For co-formulations, the entire medicine was deemed out of specification if at least one of the API was not within the 90–110% range.

Both the RDT and the Minilab-TLC are semi-quantitative devices. RDT rely on the color density of control and test lines to confirm the presence of an API at a specific concentration. The TLC portion of the Minilab kit relies on the size and migration distance of a spot that is formed by a small sample deposited on the TLC plate, compared to a good quality reference standard on the same plate.

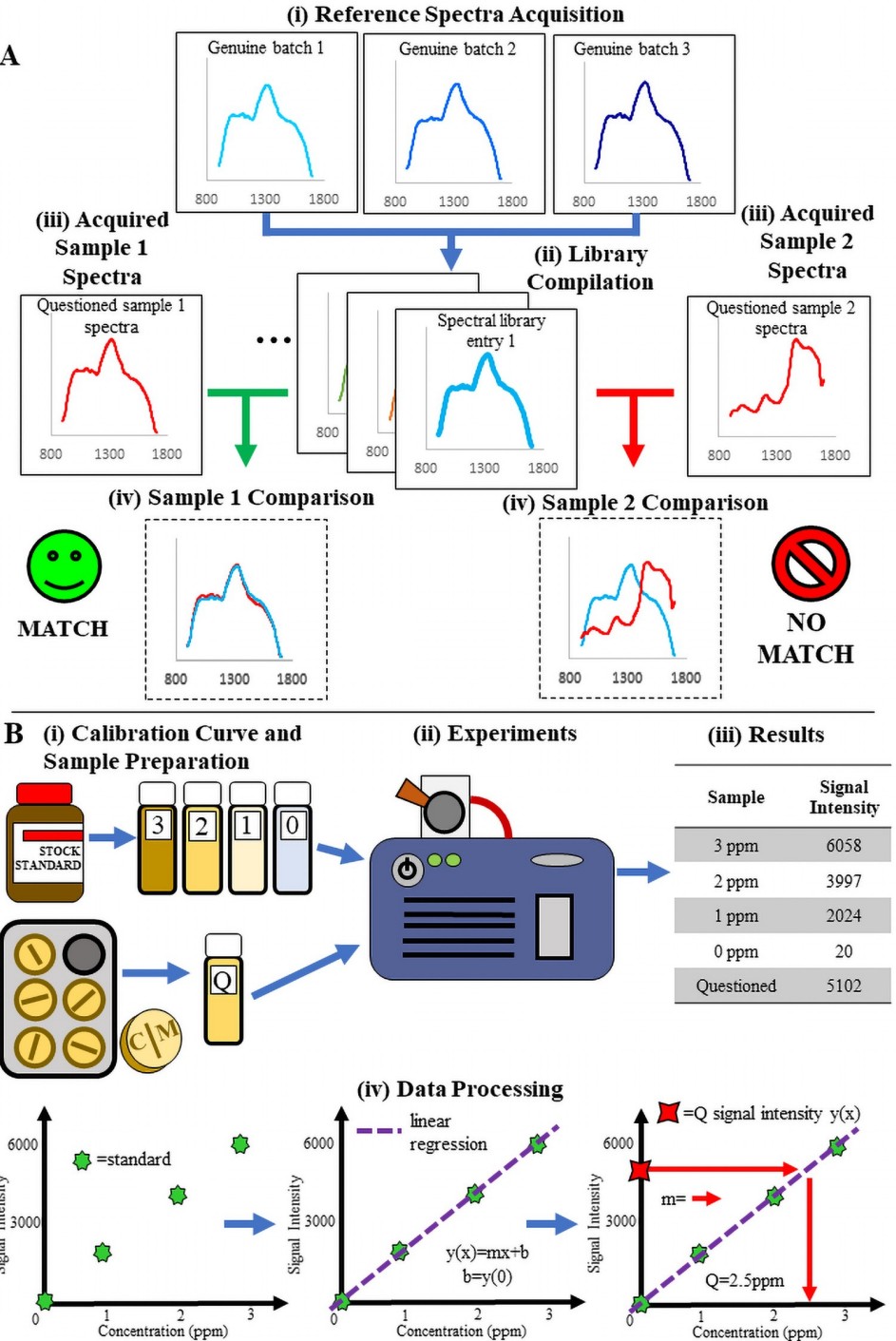

**Fig 1. Illustration of the basic processes for qualitative spectral comparison and quantitative analysis. (A)** Process for reference library creation and spectral comparison analysis. From top to bottom: (i) spectra are collected from different batches of the same medicine and compiled into a mean spectrum representative of that medicine. (ii) This mean spectrum is used to build a "library" or database that serves as the comparator against which test samples are compared. (iii) Test samples are scanned and then (iv) the test sample spectra are overlaid with the reference spectrum for visual or computational comparison to determine a pass or fail. **(B)** Illustration of a basic quantitative experiment. From left to right and top to bottom. (i) A set of standard calibration samples with increasing API concentration is prepared along with a solution of the test sample that should fit in the concentration range of those standards. (ii) All solutions are then tested on the instrument and (iii) the data collected. (iv) The data obtained is then used to build a calibration curve via linear least-squares regression. Interpolation of the peak area of the questioned sample into this curve yields the estimated API concentration.

One of the major differences between devices was whether they were specific for the API, or if they could screen the entire formulation (see S11 Appendix).

When a sample failed the first test, we operated a 'best of three' system for overall sample classification (out of the three tests performed on the failing samples, the most frequently occurring of 'pass' or 'fail' would then be the overall sample classification), in the absence of devices manufacturer's guidelines. For the single-use RDT the failing samples were rerun only once due to a limited number of tests available. For the PAD, the failing samples were rerun once, as recommended by the developer. For both the RDT and PAD a third run was conducted when the first two test results were discordant, and the 2/3 majority result was retained.

### Statistical analysis

The overall sample binary classification ('pass' or 'fail') was used to calculate the sensitivity and specificity for each instrument. Sensitivity was defined as the percentage of true positives over the total of true positives and false negatives, and specificity as the percentage of true negatives over the total of true negatives and false positives. A true positive was defined as the sample being poor quality with the device correctly giving a fail result. Sensitivity and specificity were expressed as percentages with their 95% confidence intervals (95% CI). The exact confidence interval was based on Jeffreys' confidence interval formula [25]. Sensitivities and specificities were compared by device pairs using McNemar tests. Data analysis was carried out using Microsoft Excel 2013 and STATA 14.2. The level of significance was set at p = 0.05 (two-sided).

To gauge whether the passing correlation coefficient or p-value threshold values initially set in the 4500a, MicroPHAZIR RX, Progeny, and TruScan RM spectrometers were optimal, receiver operating characteristic (ROC) curves were created. Only substandard and good quality SIM were used in the ROC curve analysis because these samples were shown to be the most challenging for the devices to distinguish. A variety of thresholds were tested to best visualize the curve and optimize the sensitivity and false positive rate (1-specificity). ROC curve analysis was also applied to the C-Vue, PharmaChk, and QDA, but the correlation coefficient or p-value thresholds were replaced with percent concentration thresholds.

## Results

All devices showed 100% sensitivity to correctly identify tablets with 0% or wrong API after removal from their packaging, except for the NIR-S-G1 that showed a sensitivity (95% CI) of 91.5% (79.6–97.6%,). Specificities of 100% were observed for most of the devices, except the C-Vue [60.0% (32.3–83.7%)], PharmaChk [50.0% (1.3–98.7%)], Progeny [95.5% (77.2–99.9%)] and the QDa [91.7% (73.0–99.0%)] (Table 2). The following subsections detail significant strengths and weaknesses observed for each during testing.

### Thin layer chromatography: Minilab

The Minilab TLC kit consisted of a hard-shell case containing all the chemical analysis equipment necessary to perform TLC experiments. It correctly characterized all good quality, no-API falsified, and wrong-API falsified FCM and SIM. Only 1 out of 21 of the SIM 50% API substandards were misidentified as good quality. A majority (16/21) of SIM 80% API substandards were misidentified as good quality as differences between the 100% and 80% reference spots on the TLC plates were difficult to distinguish visually. The Minilab is designed to detect samples with API below 80% API. The primary limitation identified was the higher requirements in terms of resources needed to conduct experiments, and the difficulties associated

**Table 2. Sensitivity and specificity to correctly determine quality of medicine samples for the 12 tested devices.**

| Devices | 0% API and wrong API samples | | Genuine Good Quality | | 50% and 80% API samples | | All poor quality samples | |
|---|---|---|---|---|---|---|---|---|
| -------- | Sensitivity (95% CI) | n | Specificity (95% CI) | n | Sensitivity (95% CI) | n | Sensitivity (95% CI) | n |
| *4500a FTIR** | 100 (93.3–100) | 53 | 100 (85.8–100) | 24 | 28.6 (15.7–44.6) | 42 | 68.4 (58.1–77.6) | 95 |
| *C-Vue* | 100 (82.4–100) | 19 | 60.0 (32.3–83.7) | 15 | 100 (81.5–100) | 18 | 100 (90.5–100) | 37 |
| *MicroPHAZIR RX** | 100 (92.5–100) | 47 | 100 (84.6–100) | 22 | 50.0 (32.9–67.1) | 36 | 78.3 (67.9–86.6) | 83 |
| *Minilab* | 100 (93.3–100) | 53 | 100 (85.8–100) | 24 | 59.5 (43.3–74.4) | 42 | 82.1 (72.9–89.2) | 95 |
| *Neospectra 2.5** | 100 (92.5–100) | 47 | 100 (84.6–100) | 22 | 5.6 (0.7–18.7) | 36 | 59.0 (47.7–69.7) | 83 |
| *NIR-S-G1** | 91.5 (79.6–97.6) | 47 | 100 (84.6–100) | 22 | 30.6 (16.3–48.1) | 36 | 65.1 (53.8–75.2) | 83 |
| *PAD* | 100 (88.8–100) | 31 | 100 (83.2–100) | 20 | 0 (0–11.6) | 30 | 50.8 (37.7–63.9) | 61 |
| *PharmaChk* | 100 (54.1–100) | 6 | 50.0 (1.3–98.7) | 2 | 83.3 (35.9–99.6) | 6 | 91.7 (61.5–99.8) | 12 |
| *Progeny** | 100 (92.5–100) | 47 | 95.5 (77.2–99.9) | 22 | 16.7 (6.4–32.8) | 36 | 63.9 (52.6–74.1) | 83 |
| *QDa* | 100 (93.3–100) | 53 | 91.7 (73.0–99.0) | 24 | 100 (91.6–100) | 42 | 100 (96.2–100) | 95 |
| *RDT* | 100 (73.5–100) | 12 | 100 (29.2–100) | 3 | 16.7 (2.1–48.4) | 12 | 58.3 (36.6–77.9) | 24 |
| *TruScan RM** | 100 (92.5–100) | 47 | 100 (84.6–100) | 22 | 22.2 (10.1–39.2) | 36 | 66.3 (55.1–76.3) | 83 |

*With substantial and upfront work, optical spectrometers could theoretically perform API quantitation. Parameters could be adjusted for better analysis of medicines containing lower-than-stated amount of API(s).

However, in this study, only default parameters provided by the manufacturer were used. It is believed, however, that potential enhancements in sensitivity and specificity could be made by optimizing threshold values and experimental settings for specific medicines.

with visual interpretation of the results. Its primary strength was the wide variety of APIs covered and its detailed instructions (see S12 Appendix).

## Single use devices: PAD and RDT

The PAD are paper card colorimetric assays with 12 testing lanes. Test samples were applied by rubbing them onto the testing lanes. The bottom of the cards are placed in water, where they develop to generate a color pattern that confirms the presence (or not) of a given API. They correctly characterized all good quality, no-API falsified, and wrong-API falsified FCM and SIM. The PAD were not designed to detect substandard medicines containing lower API amounts than stated and all 50% and 80% API substandards samples were incorrectly classified as good quality. The primary limitation of PAD was that the colors were difficult to interpret for weakly colored reaction products (S13 Appendix). Their primary strengths were that the experiments required minimal effort and consumables (see S14 Appendix).

RDT are lateral flow immunoassays that typically target a single API. They have the stated capability of distinguishing between substandard and falsified antimalarials. A subset of the RDT tested claimed to be able to detect artemether in artemether-lumefantrine co-formulated medicines. These RDT, however, were defective and not considered further. RDT specific for DHA and ART had a sensitivity of 16.7% in detecting substandards. RDT correctly characterized all the remaining good quality, no-API falsified, and wrong-API FCM and SIM. The primary limitations of RDT were the additional consumables required for their operation and the relatively extensive sample preparation needed compared to PAD. In a few cases it was difficult to read the RDT control lines on the cartridges due to inconsistent colors observed between RDT units (S15 Appendix). One advantage of RDT over PAD was their ability to detect some substandards (see S16 Appendix).

## MIR spectrometer: 4500a

The 4500a is a MIR spectrometer that performs attenuated total reflection experiments on powdered samples. It is controlled through a computer or Windows-based smartphone.

With a correlation coefficient threshold of >0.9 for pass/fail analysis, the 4500a MIR spectrometer correctly characterized all good quality, no-API falsified, and wrong-API falsified FCM and SIM. None of the 80% API substandard SIM were correctly classified as poor quality; however, over half (12/21) of the 50% API substandard SIM were correctly classified as poor quality. Primary weaknesses were its inability to test through transparent packaging because many plastics completely absorb MIR radiation, and the need to crush samples into powders. A major strength of the 4500a MIR spectrometer software was the step-by-step instructions provided to aid in correcting background, cleaning the unit, and labeling IR spectra (see S17 Appendix).

## NIR spectrometers: Neospectra 2.5, NIR-S-G1, & MicroPHAZIR RX

The Neospectra 2.5 Fourier transform NIR detector module can be used to build a device that fits a user's custom needs. It correctly characterized all good quality, no-API falsified, and wrong-API falsified FCM and SIM. All the 80% API substandards and 16 out of 21 of the 50% API substandards were misidentified as good quality samples. The primary limitation was the inability to conduct automated spectral library matching. Although users can extract the raw spectra using third-party library comparison tools, this feature was not evaluated as it was outside the study scope. Its modularity was its primary strength, allowing to configure the system for sampling both solids and liquids. Its detector had the widest NIR wavelength range in this study, allowing for more spectral information to be collected (see S18 Appendix).

The NIR-S-G1 NIR spectrometer is controlled by a smartphone via Bluetooth. Although this instrument correctly characterized all good quality medicines, it had difficulties in the analysis of the OFLO SIM. The no-API falsified, 50% substandard, and 80% substandard OFLO SIM were all incorrectly deemed as being good quality. This may have been because the limited device spectral range (Fig 2) did not reveal a sufficiently large number of significant features when comparing against the reference library entry. The library processing software was not able to detect the relatively small spectral differences observed in the ~1500 nm region (Fig 2B). Since this device was able to correctly determine that the wrong-API OFLO medicines were poor quality, the API itself may have had limited NIR spectral features distinguishable from the excipients. Additionally, one falsified SIM sample containing only starch was misidentified as good quality DHAP. The simulated DHAP sample used D-Artepp as the source of API. It is possible that if D-Artepp contained starch, this excipient may have contributed to the misclassification. Cellulose and lactose-containing falsified tablets were correctly characterized as poor quality DHAP. All other falsified samples were correctly identified. Four out of 21 (19%) of the 80% API substandards, and 10 out of 21 (48%) of the 50% API substandards were correctly identified as poor quality. The limited spectral range and limited chain-of-custody capabilities, which included the inability to record tested sample information in the software at the time of testing, were key limitations. The primary strengths were the user interface simplicity and that the NIR-S-G1 was the most portable spectrometer tested (see S19 Appendix).

The MicroPHAZIR RX handheld NIR spectrometer contains the instrument and user interface in one module. It correctly characterized all good quality, 50% API substandard, no-API falsified, and wrong-API falsified FCM and SIM. Only 1 of the 21 (4.8%) 80% API SIM substandards was correctly classified as poor quality. The primary limitation of the MicroPHAZIR RX were the instrument's bulkiness and advanced skills necessary to process the spectra to generate reference libraries, but it had an easy-to-use interface. (see S20 Appendix).

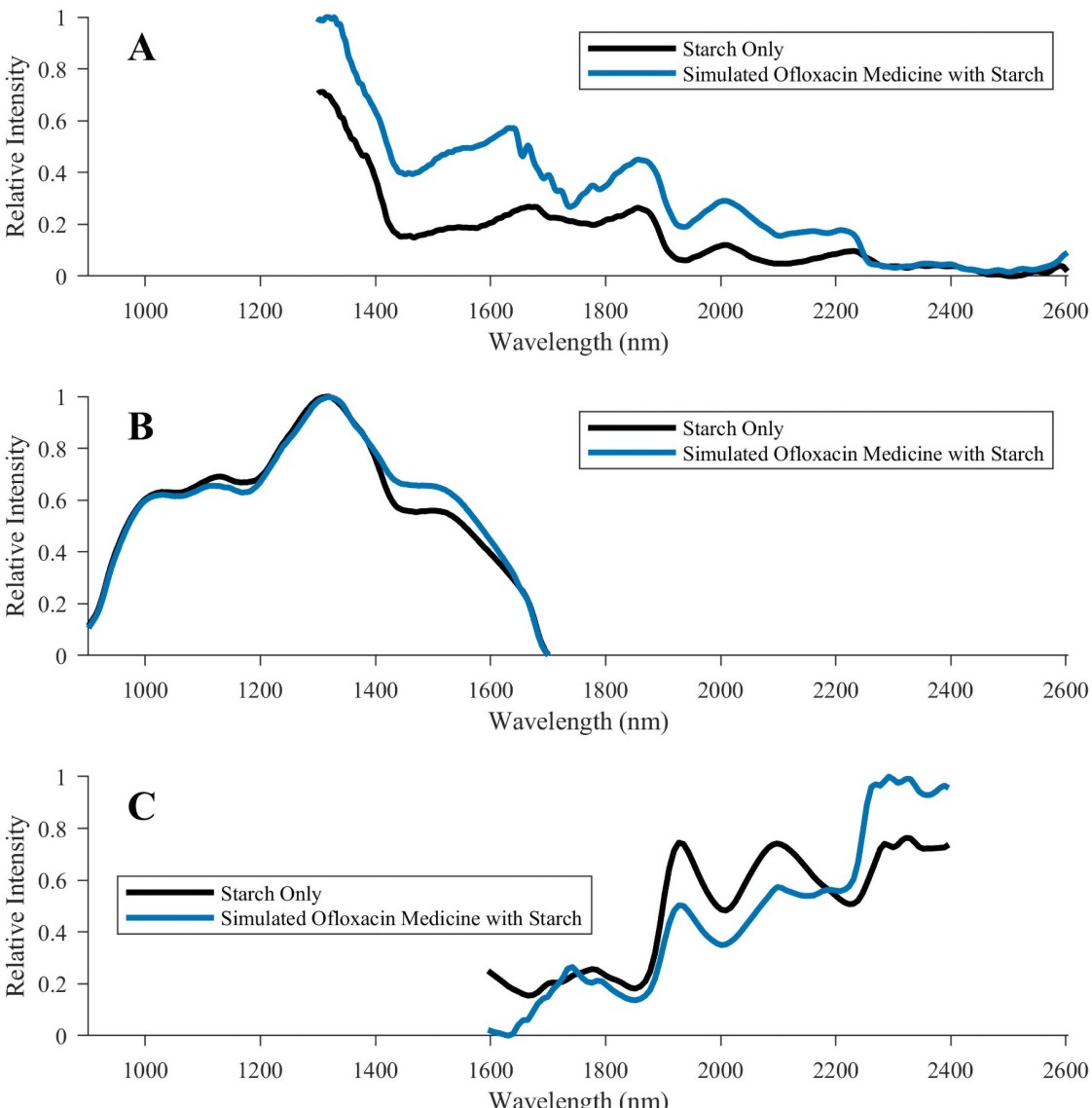

**Fig 2. Comparison of NIR spectra obtained for ofloxacin-containing simulated medicines.** Spectra were collected for ofloxacin-containing simulated medicines using the **(A)** Neospectra 2.5, **(B)** NIR-S-G1, and **(C)** MicroPHAZIR RX spectrometers. The black trace is of a falsified simulated medicine tablet containing only starch. The blue trace is of a simulated good quality ofloxacin sample that contained starch as the bulk excipient.

## Raman spectrometers: Progeny and TruScan RM

The Progeny is a handheld Raman spectrometer with a 1064 nm excitation laser, resulting in lower fluorescence than shorter wavelength lasers. It correctly characterized all the no-API falsified and wrong-API falsified FCM and SIM. All good quality samples were correctly characterized, except the FCM Augmentin (ACA), which was incorrectly determined as being Roxythroxyl (roxithromycin). The outer tablet coating of Augmentin may have been chemically similar to that of Roxythroxyl, a spectra stored in the master Progeny reference library. None of the 80% SIM substandards and only 7 out of 21 (33%) of the 50% SIM substandards were correctly characterized as poor quality. The Progeny spectrometer was heavier than other units (1.6 kg) and risked burning samples with the laser if the instrument's focal length was

not properly set-up. The primary strengths were its simple to use touch screen interface, and the lack of fluorescence signals that could overwhelm the detector (see S21 Appendix).

The TruScan RM is a handheld Raman spectrometer with a 756 nm excitation laser that correctly classified all the good quality, no-API falsified, and wrong-API falsified medicines in the FCM and SIM collections. The device only correctly classified 3 out of 21 (14%) SIM 80% API, all three being DHAP samples. Seven out of 21 (33%) SIM 50% API substandard were correctly classified. Limitations included the need for significant computer knowledge (e.g. setting up IP addresses, firewalls) to correctly complete the initial setup with the master computer and the excitation laser that caused significant fluorescence for ACA FCM and SIM samples when compared to the other non-destructive spectrometers tested (S22 Appendix). Sampling Artesun through the original sample glass vial yielded spectra that were indistinguishable from those of an empty glass vial, whereas this issue was not observed with the Progeny (Fig 3). The primary strengths of the TruScan RM were its relatively lower weight and bulkiness when compared with the Progeny and its ease of use (see S23 Appendix).

## Quantitative instruments: C-Vue, PharmaChk, & QDa

The C-Vue is a portable lightweight tabletop liquid chromatograph equipped with a manual syringe pump, a manual injector, a reverse-phase column, two detectors, and a control laptop. It was able to correctly characterize all the 80% API substandard, 50% API substandard, no-API, and wrong-API OFLO, SMTM, and ACA. The C-Vue specificity was one of the lowest in this study (60% good quality FCM and SIM misidentified as poor quality), potentially due to matrix effects from the excipients that were not used in the preparation of the calibration solutions. The lowest specificity was observed for medicines formulated with more than one API (0% and 40% for ACA and SMTM, respectively), but none of the single API good quality OFLO FCM and SIM were misclassified. Two notable limitations are its inability to detect artemisinin derivatives with the setup tested and the significant effort required to continuously ensure that the pump was pressurized by hand. Newer versions of this instrument include an upgraded mobile phase electric pump that simplifies operation. Its primary strength was the simplicity of its design, making it easy to transport, repair, or modify in the field (see S24 Appendix).

The PharmaChk is a portable microfluidic system that uses luminol-based chemiluminescence detection, contained inside a hard-shell case. The system can automatically calibrate itself and assay samples in a single experiment. It was able to correctly characterize all the 50% API substandard, no-API falsified, and wrong-API falsified ART SIM. One of the three 80% API substandards was misclassified as good quality. The FCM good quality ART was correctly classified, while the SIM 100% ART sample was incorrectly classified in two trials as poor quality, with yields of 51.6 mg (86%) and 53.75 mg (89.5%) of the stated 60 mg amount. Its primary limitations were that the prototype used could only test for ART and the reagents degraded within a few hours after being prepared. Its primary strengths were the automation of all API concentration calculations in the embedded computer system, the capability of simultaneously infusing reagent and samples, and the clear step by step instructions (see S25 Appendix).

The QDa is a benchtop single quadrupole mass spectrometer with an electrospray ion source. All the 80% API substandards, 50% API substandards, no-API falsified, and wrong-API falsified medicines were correctly classified. Two good quality FCM were misclassified as poor quality, likely due to incomplete API extraction. Limitations of the QDa were that it required significantly more extensive sample preparation than the C-Vue or the PharmaChk because it required several dilutions for the tested sample for the extracts to fall within the linear range of the instrument and not overwhelm the detector. The requirements for additional consumables such as compressed nitrogen gas, and the higher mechanical complexity of the

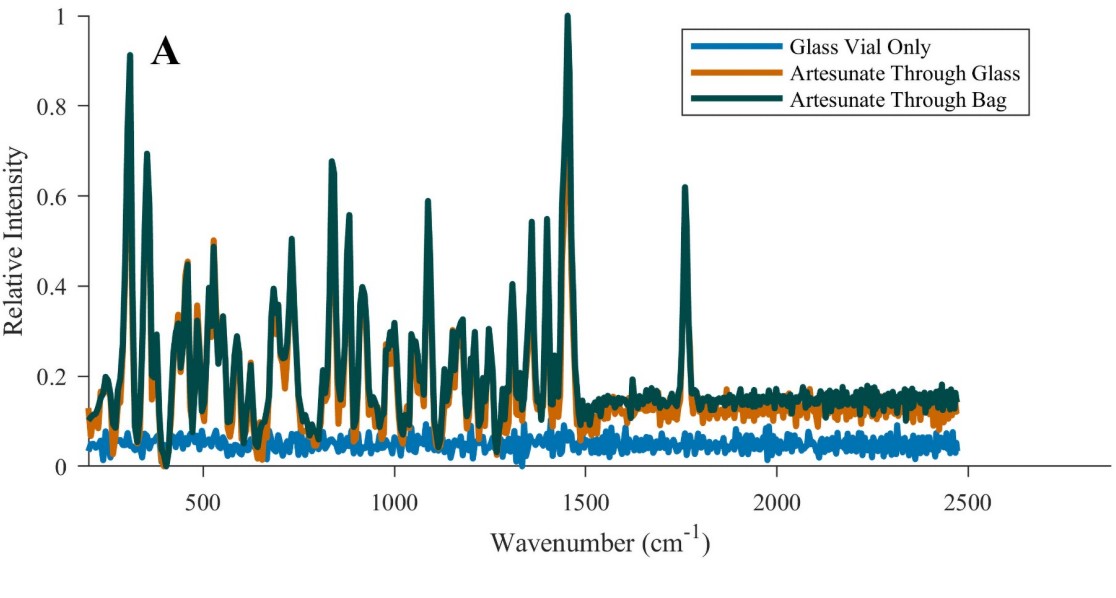

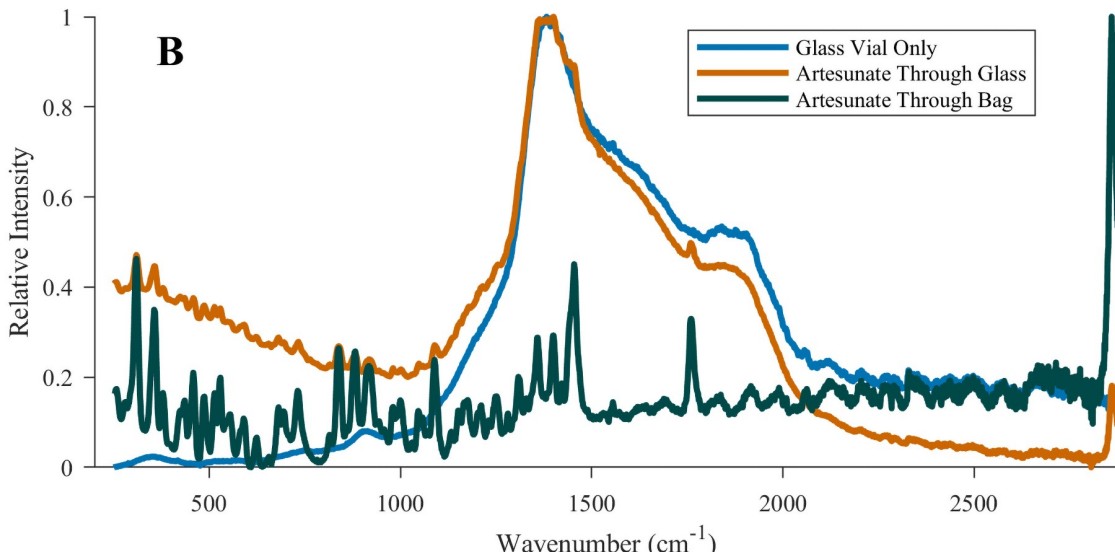

**Fig 3. Comparison of Raman spectra obtained for Artesun artesunate powder.** Raman spectra were collected with the **(A)** Progeny and **(B)** TruScan RM spectrometers for Artesun artesunate powder for injection. Spectra are provided for 1) a scan of the bottom of the Artesun glass vial containing no artesunate (blue trace), 2) a sample containing 60 mg of artesunate powder, scanned through the bottom of the glass vial (orange trace), and 3) the artesunate powder transferred to a polypropylene bag and compacted into a more localized area to enable more focused analysis (green trace).

instrument could make portability and serviceability challenging. Its primary strength was its quantitative capabilities for a wide range of APIs, and the high sample throughput once sample preparation was complete (see S26 Appendix).

## Device comparison

Paired-wise comparisons of the sensitivities showed that no device had statistically significantly lower or higher sensitivities to correctly identify 0% and wrong API samples than any other device (Table A in S27 Appendix).

Specificity of the C-Vue was significantly lower than that of all other devices except the Progeny (p = 0.0625) and the QDa (p = 0.1250) (Table B in S27 Appendix). The performances of the PharmaChk and RDT could not be compared with the C-Vue as this device was limited to test ACA, OFLO and SMTM in the present study. Because only a few genuine medicine samples were available for specificity calculations, the interpretation of statistical comparisons is limited.

The C-Vue and QDa performed better to correctly identify 50% and 80% API samples than other devices they could be compared to, except the PharmaChk (p<0.05) (Table C in S27 Appendix); the Minilab showed higher sensitivity than other devices, except the C-Vue, QDa, PharmaChk and MicroPHAZIR RX (p<0.05). The MicroPHAZIR RX showed higher sensitivity than other spectrometers (p<0.05) to correctly identify 50% and 80% API samples except the NIR-S-G1 (p = 0.0923); the Neospectra 2.5 had lower sensitivity than other spectrometers (p<0.05) except the Progeny (p = 0.2188). The PAD showed significantly lower sensitivity to correctly identify 50% and 80% API samples than the other devices except the RDT (p = 1.0) and the Neospectra 2.5 (p = 0.50) but the number of samples for comparisons was limited.

## Discussion

These results suggest that all tested devices are well-suited for detecting no API and wrong API medicines, these being common falsified medicines chemical compositions. Although detecting falsified medicines is of great public health importance, the co-detection of substandard medicines is also essential, especially as these will lead to cryptic therapeutic failure and engender antimicrobial resistance. Much needed technological improvements, as revealed by this study, are the ability to quantitate APIs *via* Raman/IR spectrometers or disposable devices, and improvement of the specificity of some of the quantitative devices. Although all substandards were successfully detected with quantitative devices, except two ART samples with the PharmaChk, their less-than-optimal specificity is of concern. Reduced specificities increase the number of confirmatory testing required in quality control laboratories, resulting in increasing cost and loss of time and resources. Quarantine or withdrawal of the samples failing the screening technologies from the market due to device limitations can also negatively impact manufacturers, pharmacies, and ultimately patients.

Many FCM tablets tested had coatings that made sampling of the tablet bulk difficult by spectroscopic techniques. To illustrate this point, the spectrometers (except the 4500a FTIR that cannot scan intact tablets) were used to sample AMK 1000 mg (ACA) tablets, both intact and crushed. The corresponding overlaid spectra shown in S22 Appendix indicate clear differences between the spectra of coated and crushed medicines. The spectral fingerprint of the outer coating of intact tablets, depending on coating composition and thickness, may be insufficient to derive the quality of the medicine itself as a whole. Reference library creation could also be affected if the operator sampled the coating but attempted to match the spectrum to that of a crushed tablet. We were unable to find evidence on the consequences of different coatings and their thickness on spectra [3]. S28 Appendix summarizes the perceived difficulty with testing different medicine formulations, such as capsules, water based medicines, powders, and creams/gels with each device.

Optimization of the spectrometers' software's pass/fail threshold values in performing database matching could boost the instruments' sensitivities and specificities. The ROC curves constructed in Fig 4 for SIM with the 4500a, MicroPHAZIR RX, Progeny, and TruScan RM spectrometers show that setting a higher threshold improve devices sensitivities, but at the cost of reduced specificity. The quantitative devices' precision could also be optimized by adjusting the concentration threshold used for pass/fail decisions. Lowering the pass threshold

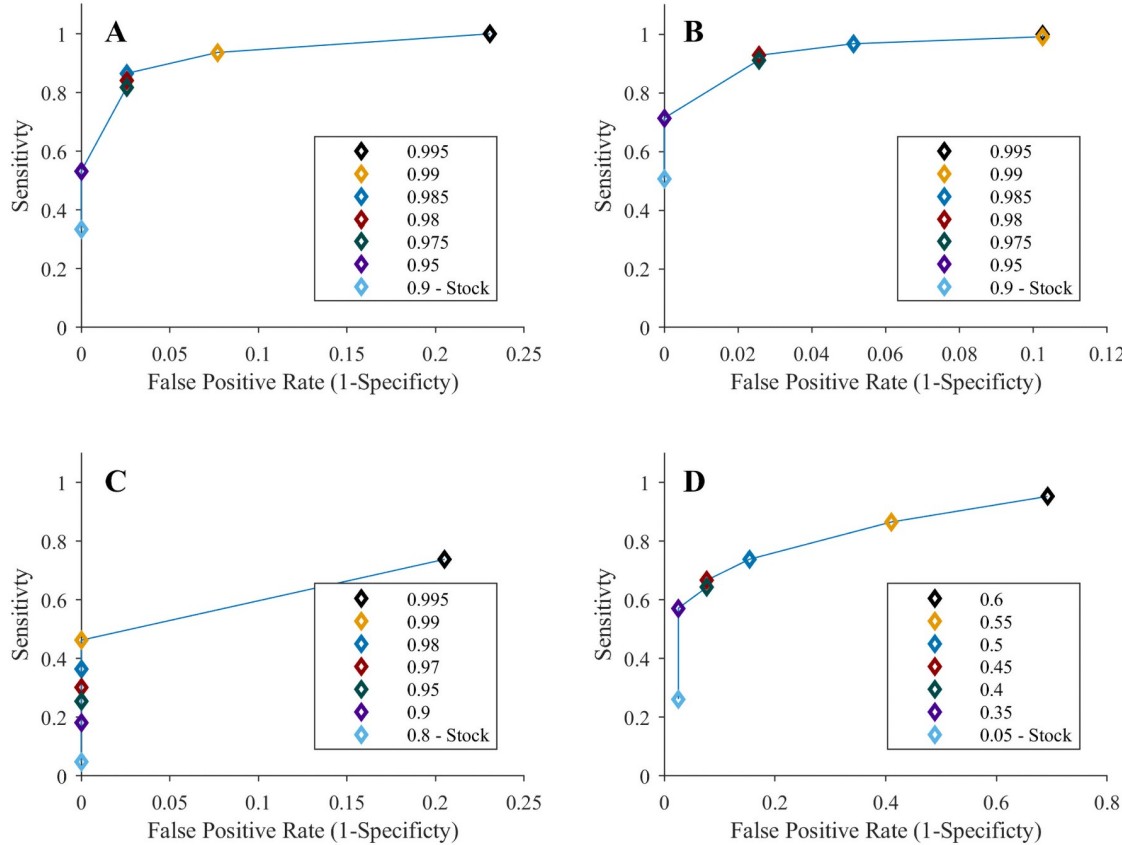

**Fig 4. Receiver operating characteristic (ROC) curves for substandard analysis with the spectrometers.** ROC curves were created for the **(A)** 4500a, **(B)** MicroPHAZIR RX, **(C)** Progeny, and **(D)** TruScan RM spectrometers. ROC curves were based only on the results for simulated substandard and good quality medicines. Each legend identifies the threshold chosen for each point, with the one labelled "Stock" being the threshold used for the study. The stock thresholds for the MicroPHAZIR RX's correlation coefficient, Progeny's correlation coefficient, and TruScan RM's p-value were the default values set by the manufacturer. The 4500a stock threshold was selected for the study since that instrument did not output pass/fail results.

concentrations for the quantitative devices (C-Vue, PharmaChk, & QDa) improves their specificities, but at the cost of reduced sensitivity (S29 Appendix). Therefore, the pass/fail threshold values for these instruments must be balanced to optimize the reliability of medicine quality classification. In most circumstances public health risks suggest that optimizing sensitivity for detecting SF medicines will be more important than specificity.

Limitations of our study include: the limited number of APIs tested; difficulties with reference libraries creation for spectrometers; the limited comparability of reference libraries due to the variability of spectrometers; the investigators not being blinded to sample identity; the limited optimization of the devices to allow API quantitation; and the low power to compare sensitivities and specificities between devices (see S30 Appendix for more details).

With additional development and optimization, many of these devices may be able to perform both quantitative and qualitative analysis. For example, methods could be developed for spectrometers to perform quantitative analyses by selecting a spectral feature correlated to a specific chemical bond in an API, or by monitoring changes in the entire spectrum and correlate those to the differences in API concentration. A key obstacle to portable spectrometer quantitation would be instrument calibration per brand because they analyze the entire formulation that will vary between different excipient recipes. To ensure a reliable

calibration at a variety of different API concentrations, the calibrants should be chemically similar to the medicine of interest due to potential matrix effects. The QDa mass spectrometer could be adapted for qualitative analysis by allowing the instrument to record the entire mass spectrum to analyze as many ingredients in the formulation as possible. However, analyzing a wider mass range risks less accurate quantitation compared to single ion monitoring used in this study.

Which device is optimal for different positions within the medicine distribution system will depend on the question being asked. For example, what type of medicine or API is tested, whether qualitative or quantitative assays are required, and what human and financial capacity is available. The Minilab, PAD, and RDT require minimal training and infrastructure to implement in the field rapidly. However, the PAD and RDT are limited due to narrow range of detected API. The spectrometers have the advantages of being non-destructive (except the 4500a MIR spectrometer) and can analyze a broader API diversity. The spectrometers suffer from issues with tablet coatings, the spectral range of the NIR-S-G1 spectrometer, and fluorescence issues with the TruScan RM can reduce accuracy for some medicines. The smartphone simplicity with the NIR-S-G1 is highly desirable. With the increasing global prevalence of smartphone technology, connectivity using cell phone networks, and Wi-Fi to constantly update reference libraries, software, and back-up data could ensure these devices are performing to the best of their ability with rapid access to troubleshooting [26]. The simplicity of the C-Vue liquid chromatographer, the automation of the PharmaChk microfluidic system, and the sample variety and accuracy for the QDa mass spectrometer are all desirable traits for an optimal portable confirmatory instrument.

The following devices were selected for field testing described in the field evaluation in Laos—third paper of the PLOS NTD's collection: 4500a, Minilab, MicroPHAZIR RX, NIR-S-G1, PAD, Progeny, and TruScan RM. The main reasons were their ease of use, ease of training, portability, minimal consumables, and ease of export. The RDT were selected as field-suitable but could not be included in the field evaluation because not enough tests were available. The quantitative devices were not selected because of the consumables and training requirements. In addition, the QDa had high mechanical complexity, making export and setup difficult. The quantitative devices are highly capable, but the consumables and mechanical complexity make transport difficult and the devices would be more appropriate for a laboratory.

## Supporting information

**S1 Appendix. Brief technology description.**
(PDF)

**S2 Appendix. Notable device configurations and constraints.**
(PDF)

**S3 Appendix. Photographs of the devices.**
(PDF)

**S4 Appendix. Device descriptions.**
(PDF)

**S5 Appendix. Device protocols.**
(PDF)

**S6 Appendix. Field collected medicines tested and UPLC results.**
(PDF)

**S7 Appendix. Details about simulated medicine preparation.**
(PDF)

**S8 Appendix. Brief samples handling protocols.**
(PDF)

**S9 Appendix. Reference library creation questionnaires.**
(PDF)

**S10 Appendix. US, International, Chinese, and British pharmacopeial standards for the studied APIs.**
(PDF)

**S11 Appendix. API vs formulation screening.**
(PDF)

**S12 Appendix. Minilab thin layer chromatography kit results.**
(PDF)

**S13 Appendix. PAD color analysis.**
(PDF)

**S14 Appendix. Paper Analytical Devices (PAD) results.**
(PDF)

**S15 Appendix. RDT analysis.**
(PDF)

**S16 Appendix. Rapid Diagnostic Test (RDT) results.**
(PDF)

**S17 Appendix. 4500a single reflection FTIR spectrometer results.**
(PDF)

**S18 Appendix. Neospectra 2.5 NIR spectrometer results.**
(PDF)

**S19 Appendix. NIR-G-G1 spectrometer results.**
(PDF)

**S20 Appendix. MicroPHAZIR RX NIR spectrometer results.**
(PDF)

**S21 Appendix. Progeny Raman spectrometer results.**
(PDF)

**S22 Appendix. Intact vs. crushed ACA tablet analysis by Raman and NIR spectrometers.**
(PDF)

**S23 Appendix. Truscan RM Raman spectrometer results.**
(PDF)

**S24 Appendix. C-Vue liquid chromatograph results.**
(PDF)

**S25 Appendix. PharmaChk microfluidic device results.**
(PDF)

**S26 Appendix. QDa mass spectrometer results.**
(PDF)

**S27 Appendix. Pair-wise analysis.**
(PDF)

**S28 Appendix. Table describing the degree of difficulty to analyze other, less common, medicine formulations relative to the analysis of a tablet.**
(PDF)

**S29 Appendix. Receiver operating characteristic (ROC) curves for the quantitative devices.**
(PDF)

**S30 Appendix. Limitations of the study.**
(PDF)

## Acknowledgments

We acknowledge invaluable support from Vayouly Vidhamaly, Kem Boutsamay and Phonepasith Boupha in compiling data for data tables. We also acknowledge William Griggers for help with MiniLab testing. We are very grateful to Bounthaphany Bounxouei, past Director of Mahosot Hospital; to Bounnack Saysanasongkham, Director of Department of Health Care, Ministry of Health; to H.E. Bounkong Syhavong, Minister of Health, Lao PDR; to the Director, Dr Manivanh Vongsouvath, and staff of the Microbiology Laboratory and LOMWRU for their help and support. We are very grateful for the support of ADB, especially Dr Susann Roth, Dr Sonalini Khetrapal, Editha S. Santos and ADB Consultant Dr Douglas Ball.

## Author Contributions

**Conceptualization:** Stephen C. Zambrzycki, Celine Caillet, Laura C. Geben, Matthew C. Bernier, Paul N. Newton, Facundo M. Fernández.

**Data curation:** Stephen C. Zambrzycki, Celine Caillet, Serena Vickers, David V. Donndelinger, Paul N. Newton.

**Formal analysis:** Stephen C. Zambrzycki, Celine Caillet, Serena Vickers, Marcos Bouza, David V. Donndelinger, Paul N. Newton.

**Funding acquisition:** Paul N. Newton, Facundo M. Fernández.

**Investigation:** Stephen C. Zambrzycki, Celine Caillet, Serena Vickers, Marcos Bouza, David V. Donndelinger, Matthew C. Bernier, Paul N. Newton.

**Methodology:** Stephen C. Zambrzycki, Celine Caillet, Serena Vickers, Laura C. Geben, Matthew C. Bernier, Paul N. Newton, Facundo M. Fernández.

**Project administration:** Celine Caillet, Serena Vickers, Paul N. Newton, Facundo M. Fernández.

**Supervision:** Celine Caillet, Serena Vickers, Paul N. Newton, Facundo M. Fernández.

**Writing – original draft:** Stephen C. Zambrzycki, Facundo M. Fernández.

**Writing – review & editing:** Stephen C. Zambrzycki, Celine Caillet, Serena Vickers, Marcos Bouza, David V. Donndelinger, Laura C. Geben, Matthew C. Bernier, Paul N. Newton, Facundo M. Fernández.

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
