## [Decision Letter · Decision Letter 0]

1 Mar 2021

Dear Prof. Fernandez,

Thank you very much for submitting your manuscript "Laboratory evaluation of twelve portable devices for medicine quality screening." for consideration at PLOS Neglected Tropical Diseases. As with all papers reviewed by the journal, your manuscript was reviewed by members of the editorial board and by several independent reviewers. The reviewers appreciated the attention to an important topic. Based on the reviews, we are likely to accept this manuscript for publication, providing that you modify the manuscript according to the review recommendations. 

Sincerely,

Subash Babu

Associate Editor

Benjamin Althouse

Deputy Editor

Reviewer's Responses to Questions

**Key Review Criteria Required for Acceptance?**

**Methods**

-Are the objectives of the study clearly articulated with a clear testable hypothesis stated?

-Is the study design appropriate to address the stated objectives?

-Is the population clearly described and appropriate for the hypothesis being tested?

-Is the sample size sufficient to ensure adequate power to address the hypothesis being tested?

-Were correct statistical analysis used to support conclusions?

-Are there concerns about ethical or regulatory requirements being met?

Reviewer #1: Suggest authors to elaborate some part of the study methodology as suggested in the main text in the document, e.g. the preparation of the simulated medicines formulation.

Reviewer #2: (No Response)

**Results**

-Does the analysis presented match the analysis plan?

-Are the results clearly and completely presented?

-Are the figures (Tables, Images) of sufficient quality for clarity?

Reviewer #1: The use of terms should be clearly defined as commented in the main text.

Reviewer #2: (No Response)

**Conclusions**

-Are the conclusions supported by the data presented?

-Are the limitations of analysis clearly described?

-Do the authors discuss how these data can be helpful to advance our understanding of the topic under study?

-Is public health relevance addressed?

Reviewer #1: This article would help clarify some of the questions and challenges many country medicines regulatory authorities in low and middle-incomes countries are facing in understanding the pros and cons of the existing field detection technologies to are being considered to introduce/deployed for combating the substandard and falsified medicines in their country.

Reviewer #2: (No Response)

**Editorial and Data Presentation Modifications?**

Reviewer #1: (No Response)

Reviewer #2: (No Response)

**Summary and General Comments**

Reviewer #1: Support the publication

Reviewer #2: 1. Well written manuscript evaluating portable devices for medicine quality screening.

2. To include cost of device, approximate cost and time duration in performing one test in TABLE-1.

3. In Discussion, to include cost effectiveness of each testing method.

4. In Discussion, to discuss on performance of device in point-of-care or field settings. If some devices needs laboratory facility or temperature needs to be maintained, do make a mention about in limitations section.

PLOS authors have the option to publish the peer review history of their article (what does this mean?). If published, this will include your full peer review and any attached files.

Reviewer #1: No

Reviewer #2: Yes: Syed Hissar

Figure Files:

Data Requirements:

Reproducibility:

References

---

## [Editor Report · Decision Letter 1]

2 Apr 2021

Dear Prof. Fernandez,

We are pleased to inform you that your manuscript 'Laboratory evaluation of twelve portable devices for medicine quality screening.' has been provisionally accepted for publication in PLOS Neglected Tropical Diseases.

Best regards,

Subash Babu

Associate Editor

Benjamin Althouse

Deputy Editor

---

## [Editor Report · Acceptance letter]

23 Aug 2021

Dear Prof. Fernández,

We are delighted to inform you that your manuscript, "Laboratory evaluation of twelve portable devices for medicine quality screening.," has been formally accepted for publication in PLOS Neglected Tropical Diseases.

Best regards,

Shaden Kamhawi

co-Editor-in-Chief

Paul Brindley

co-Editor-in-Chief
